# Is a PCSK9 Inhibitor Right for Your Patient? A Review of Treatment Data for Individualized Therapy

**DOI:** 10.3390/ijerph192416899

**Published:** 2022-12-16

**Authors:** Roman A. Beltran, Kyle J. Zemeir, Chase R. Kimberling, Mary S. Kneer, Michelle D. Mifflin, Tom L. Broderick

**Affiliations:** 1Department of Biomedical Sciences, College of Graduate Studies, Midwestern University, Glendale, AZ 85308, USA; 2College of Osteopathic Medicine, Midwestern University, Glendale, AZ 85308, USA; 3Laboratory of Diabetes and Exercise Metabolism, Department of Physiology, College of Graduate Studies, Midwestern University, Glendale, AZ 85308, USA

**Keywords:** 3-hydroxy-3-methylglutaryl-coenzyme A (HMG-CoA), cardiovascular, hyperlipidemia, low-density lipoprotein, proprotein convertase subtilsin-kixen type 9 inhibitor (PCSK9i), statin-associated muscle symptoms (SAMS), statin intolerance, statin resistance

## Abstract

Introduction: In the United States, a significant amount of the population is affected by hyperlipidemia, which is associated with increased levels of serum low-density lipoprotein (LDL-C) and risk of cardiovascular disease. As of 2019, the guidelines set by the American College of Cardiology/American Heart Association advocate for the use of statins as the major contributor to lowering serum LDL-C. While proven to be effective, side effects, including muscle-related symptoms and new-onset diabetes mellitus, can make patients unable to tolerate statin therapy. Additionally, there is a subset of the population which does not approach a recommended LDL-C goal on statin treatment. Due to these findings, it was deemed necessary to review the literature of current statin-alternative lipid-lowering therapies. Methods: A systematic review of preclinical and clinical papers, and a current meta-analysis, was performed using PubMed and Google Scholar. Following the literature review, a meta-analysis was conducted using ProMeta 3. Results: Through systematic review and meta-analysis of the current literature, it is suggested that newer lipid-lowering therapies such as proprotein convertase subtilsin-kixen type 9 (PCSK9) inhibitors are a safe and effective statin alternative for the population with statin intolerance. PCSK9 inhibitors were shown to have no significant effect in causing myalgia in patients and showed no increase in adverse cardiovascular outcomes compared to a control of a current antilipemic medication regimen. Discussion: There are many statin-alternative therapies that should be investigated further as a potential replacement for patients with statin intolerance or as an addition for patients with statin resistance.

## 1. Introduction

Hypercholesterolemia is a widespread disease in American society. It is estimated that more than 100 million people in the United States, corresponding to roughly 53 percent of adults, have elevated low-density lipoprotein-C (LDL-C) [1]. According to clinical practice guidelines, of the 100 million adults with increased LDL, only 50% of patients receive treatment and 17.5% of these patients fail to manage their condition adequately [1]. Without proper management, the risk of developing atherosclerotic cardiovascular disease (ASCVD) increases for these patients.

As of 2019, the guidelines set by the American College of Cardiology/American Heart Association (ACC/AHA) advocate for the use of a class of drugs called 3-hydroxy-3-methylglutaryl-coenzyme A (HMG-CoA) inhibitors, as the primary drug for lowering serum LDL-C. Common statins include atorvastatin (Lipitor^®^), rosuvastatin (Crestor^®^), pitavastatin (Livalo^®^), and pravastatin (Pravachol^®^). The ACC/AHA recommends statin use for individuals between the ages of 40 and 75 who have one or more risk factors for cardiovascular disease (CVD), or have greater than 10 percent risk of a cardiovascular event (CVE) within the next ten years [2].

Statins lower LDL-C by competitively inhibiting HMG-CoA reductase, the rate-limiting step in the cholesterol biosynthetic pathway (Figure 1) [2]. Their mechanism of action occurs primarily in the liver. Statins increase LDL receptors in the hepatocyte membrane, increasing reuptake of LDL particles, thus resulting in decreased endogenous hepatic cholesterol synthesis. Statins also increase high-density lipoprotein-C (HDL-C) and slightly decrease plasma triglycerides (TG). The decrease in triglycerides is achieved via the upregulation of the LDL receptors and the internalization of LDL-containing triglycerides [3]. Additionally, the inhibition of cholesteryl ester transfer protein (CETP) results in less transfer of cholesterol esters from HDL-C to very-low-density lipoproteins (VLDL). This increases the amount of HDL-C and lowers VLDL [4]. Furthermore, statins have been shown to stabilize atherosclerotic plaques, thereby lowering the risk of cardiovascular morbidity and mortality. In addition to lowering lipids, statin medications are also associated with lowering the risk of many cancers including breast, colorectal and prostate cancer, three of the most common cancers affecting men and women [5]. Simvastatin is known to upregulate cell cycle regulators p53 and p38MAPk via phosphorylation and acylation in colorectal cells, leading to apoptosis in colorectal cell lines [6]. While statins are not first line as an antineoplastic agent, this quality is worthy of mention. Finally, statins increase PCSK9 and LDLR transcription through a cholesterol-dependent mechanism, with PCSK9 transcriptional activity increase greater than that of the LDLR [7]. These data highlight the multifaceted regulation of cholesterol, thus requiring knowledge of the entire pathway and the different pharmaceuticals available to treat hyperlipidemia.

Due to the proposed associations with various side effects, some patients are still reluctant or unable to take statins. These adverse events may include musculoskeletal pain (myalgia), new-onset diabetes (NODM), respiratory infections, gastrointestinal events, and headaches [1]. As frequently discussed, an adverse event of statin medication is myalgia. Approximately 10–20 percent of patients on statin therapy experience muscle-related symptoms such as myalgia, weakness, and inflammation [1]. It has been suggested that statin therapy may be correlated with an increased severity of heart failure (HF) in patients with a history of ischemia and that there may be an increased mortality in HF patients treated with statins [8,9]. Studies have purported that statin therapy temporarily elevates blood glucose readings in patients at risk of developing diabetes, leading patients to believe statin therapy causes NODM [10,11,12,13]. Multiple mechanisms have been proposed regarding how statins elevate blood sugars, but most are related to temporarily increased insulin resistance or impaired insulin secretion [14]. These factors result in patients being unable to tolerate or unwilling to resume statin therapy and, therefore, requiring non-statin treatment to control cholesterol levels and prevent CVEs. Moreover, a significant number, 10–20 percent, of patients fail to respond to treatment [1].

There are several novel statin alternatives, including PCSK9 inhibitors, ezetimibe, bempedoic acid, red yeast rice, and inclisiran. In this review, these options were explored, with a focus on PCSK9 inhibitors. Nine meta-analyses were chosen to complete a comprehensive systematic review. The purpose of the review was to determine the potential of PCSK9 inhibitors as an alternative to traditional statin therapy. In addition, the feasibility and limitations of clinical applications are discussed for these new classes of antilipemic drugs. In this paper, patients who cannot tolerate statins due to adverse effects after trying two different statins at differing doses are defined as having “statin intolerance,” and those whose LDL-C levels remain elevated despite optimal therapy are defined as having “statin-resistant hyperlipidemia”.

## 2. Background

### 2.1. Statin Association with Myopathy

Although statins are generally considered to be well tolerated, there are significant adverse events that can lead to intolerance and discontinuation of treatment. The most common of these is myopathy, with mild muscle symptoms being reported in 10–20 percent of the patients treated with statins (evolocumab, 140 mg biweekly or 420 mg monthly) in large community-based studies [15]. While ‘myopathy’ refers to a wide range of muscle-related symptoms, ‘myositis’ is defined as the increase in creatine kinase (CK) levels. Rhabdomyolysis is the most severe form of myositis, characterized by excessive CK elevation resulting from massive muscle destruction and myoglobinuria. Rhabdomyolysis is only reported in 0.003–0.1% of statin-treated patients [16].

There are several proposed mechanisms for statin-induced myopathy. The *SLCO1B1* gene locus occupies 109 kb on chromosome 12 (Chr 12p12.2). The common c.521T > C variant rs4149056 has been associated with decreased clearance for a number of drugs in vivo. This variant is the most widely recognized as contributing significantly to statin myopathy [17]. All statins are transported by at least one of the known adenosine triphosphate (ATP)–binding cassette and solute carrier (SLC) transporters. Multiple isoforms of the cytochrome P450 (CYP450) enzymes are involved in the oxidative metabolism of statins, with many also undergoing conjugation via uridine 5′-diphospho-glucuronosyltransferase (UGT) enzymes. Variations in an individual enzyme’s functions seem to have little effect on statin metabolism, suggesting that these configurations in combination account for the variation in response and myopathy reported by statin users [18]. Replication studies in a small population have identified three SNPs, rs9342288, rs1337512, and rs3857532, in the eyes shut homolog *(EYS)* on chromosome 6, suggestive of an association with the risk of severe statin myopathy [19,20].

Statin-treated patients have lower cholesterol synthesis and production of prenylated proteins and dolichols, which help with cell maintenance and chromatin organization and protein N-glycosylation, respectively [16]. This can lead to membranolysis and/or apoptosis in skeletal muscle. Since cholesterol is a basic component of the cell membrane and contributes to its stability, the statin reduction of cholesterol can result in changes in membrane fluidity. Membrane alterations can modulate the function of sodium, potassium, and chloride channels, subsequently damaging myocytes [16]. Statins also inhibit other intermediates in the mevalonate pathway such as farnesyl pyrophosphate and geranylgeranyl pyrophosphate, which are involved in post-translational modification of proteins such as small guanosine triphosphatases (GTPases) and lamins. These prenylated proteins are important for cell maintenance and chromatin organization. Dysprenylation of small GTPases has been shown to cause muscle fiber degeneration and apoptosis. Lamin dysprenylation may result in fragile nuclear membranes and cause cell apoptosis. Dolichols, another intermediate in the mevalonate pathway, promote proteins’ N-glycosylation. The inhibition of dolichol production by statins impedes the expression of receptors and production of structural proteins, which may also result in myopathy [16].

Several studies report a decrease in plasma coenzyme Q10 (CoQ10) levels in statin-induced myopathy. CoQ10 shuttles electrons between complex I and complex II in the electron transport chain. This decreased concentration of CoQ10 has been observed not only in circulation, but also in skeletal muscle of statin-treated humans and rodents [21]. Decreased CoQ10 levels have been associated with impaired mitochondrial function and elevated ROS as well as inhibition of the electron transport chain and β-oxidation. Impaired mitochondrial function and elevated reactive oxygen species (ROS) can also result from reduced CoQ10, as well as inhibition of the electron transport chain and β-oxidation [16]. Accelerated mitochondrial permeability transition (MPT) pore opening can lead to increased intracellular calcium levels, which can cause cellular apoptosis. Protein Kinase C (PKC) inhibition of voltage-dependent chloride (CLC-1) channels with consecutive hyperexcitability of the cytoplasmic membrane can also lead to apoptosis [16].

Supplementation with CoQ10 to treat statin-induced myopathy has shown contradictory results. A study conducted by Derosa et al. [21] found a statistically significant decrease in asthenia, myalgia, and muscle pain in patients taking CoQ10 with either lovastatin, pravastatin, simvastatin, atorvastatin, or rosuvastatin (10–40 mg) compared to placebo. These results are in line with a similar meta-analysis published earlier by Qu et al. [22]. Currently, there are no clinical guidelines recommending CoQ10 supplementation to promote adherence to statin therapy.

Impaired calcium signaling may also contribute to statin-induced myopathy. Statins induce mitochondrial depolarization and calcium release, resulting in calcium waves and subsequent calcium release by the sarcoplasmic reticulum. This is a proposed mechanism of caspase activation and apoptosis. This increase in calcium may also increase phospholipid-dependent PKC activity, promoting inhibition of CLC-1 channels with consecutive hyperexcitability of the cytoplasmic membrane. This has been shown in rats chronically treated with atorvastatin or fluvastatin (Figure 2) [16,23].

Vitamin D is an important modulator of calcium metabolism. Since the 1970s, it has been common knowledge that vitamin D deficiency, even in the absence of statin use, can predispose patients to myalgias and poor muscle function. Serum levels below 30 nmol/L (12 ng/mL) are associated with severely impaired muscle function and decreased muscle strength [24,25]. The prevalence of vitamin D deficiency in statin-induced myalgias has been reported. Statin-induced myalgias were resolved by correcting serum vitamin D (expressed as 1,25(OH)_2_D) levels [26,27,28]. These findings were later confirmed in a study involving patients with intolerance to two or more statins because of myalgia, myositis, myopathy, or myonecrosis, who were found to have low (<32 ng/mL) serum 1,25(OH)_2_D levels. After correction to 50–80 ng/mL, statins were reintroduced in these patients and titrated to bring LDL-C to below the Adult Treatment Panel (ATP) III guidelines. After a period of 24 months, 78 of the 146 patients remained myalgia-free [29]. Further studies and meta-analyses continue to suggest similar associations [30].

### 2.2. Statins and Myocardium

With knowledge of the reported prevalence of statin-associated muscle symptoms (SAMS), it is reasonable to raise concern about the health of the myocardium while on statin medication. HF is the inability of the heart to perfuse the tissues of the body and can be categorized into ischemic and non-ischemic HF [31]. Statins help prevent ischemic HF by delaying atherosclerotic plaque development. However, one must also be cautious of their use in patients with pre-existing non-ischemic HF. Non-ischemic HF has a multifactorial etiology and leads to substantial global comorbidities [31]. HF is classified based on ejection fraction (EF) as preserved EF (HFpEF), reduced EF (HFrEF), mildly reduced (HFmrEF), and improved EF (HFimpEF) [32]. A study in 2006 by Sola et al. [33] demonstrated that in patients with non-ischemic HF, left ventricular EF is increased with decreases in pro-inflammatory mediators interleukin (IL-6), tumor-necrosis factor-alpha receptor type II (TNF-aRII), and C-reactive protein (CRP) while on a daily dose of 20 mg atorvastatin. Additionally, subjects had an increased EF and an attenuation in left-ventricular remodeling. These data, in conjunction with the fact that simvastatin (10 mg daily) promotes nitric oxide synthesis in endothelial tissue and flow-mediated vasodilation, suggest that in myocardium not affected by ischemia, statins may be cardioprotective [34]. However, once ischemia has occurred and a patient has developed some degree of HF, there is debate on the role of statins in HF. While statins do decrease the risk of developing ischemic heart disease by lowering the rates of atherosclerosis, current guidelines do not advocate for initiating new HF patients on a statin, as the data do not show a significant decrease in mortality [8]. A longitudinal study by Charach et al. [9] found that in patients under 70 years of age, a lower LDL-C level was correlated with increased mortality, specifically in HF patients treated with statins. Increased mortality was noted in subjects with ischemic, non-ischemic, HFpEF, and HFrEF. It was suggested that LDL particles can neutralize bacterial endotoxins and lower levels of IL-6, TNF-α, CRP, and other pro-inflammatory molecules, and lower LDL-C levels would decrease its buffering ability to protect against damage to the vascular endothelium [9]. This conflicts with another study where cardiac troponin was measured in over 3000 men with moderate hypercholesterolemia and no prior MI over 5 years, and there was a significant decrease in troponin measured in subjects taking 40 mg pravastatin daily [35]. These data would suggest a possible benefit to the myocardium as cardiac troponin is a potent marker for myocardial injury. Currently, the recommendation for patients who have HF and a clinical indication for lipid-lowering therapy is to either continue with statin treatment or start patients who were not previously on one on a low-dose statin [36]. Among patients with systolic heart failure, lipid-lowering therapy with rosuvastatin was not associated with a reduction in the primary endpoint of CV death, MI, or stroke at a median follow-up of 32.8 months compared with placebo. Despite effectively reducing LDL, triglycerides, and CRP with rosuvastatin, there was no impact on clinical events, other than CV hospitalizations. These findings were observed despite the high levels of prior coronary disease in the population. The authors noted that low total cholesterol levels in this population have actually been associated with worse outcomes in prior studies. While there was no significant benefit on clinical events, except reduction in hospitalization for CV causes and heart failure, there was also no evidence of harm associated with the reduction in LDL levels in the present study of older patients with relatively severe systolic dysfunction [37].

### 2.3. Statin Association with Diabetes

Studies have shown that statin therapy appears to unveil NODM in certain patient populations, which is a vital concern and a motivating factor for evaluating alternative treatments. Multiple mechanisms have been proposed regarding how this occurs, but most are related to increased insulin resistance or impaired insulin secretion [14]. To date, it has not yet been determined whether this is primarily due to intracellular depletion of cholesterol or the direct effect of statins. Since statins inhibit the mevalonate pathway, this causes reductions in other downstream products. The major components that might be involved in NODM are geranylgeranyl pyrophosphate (GGPP), farnesyl pyrophosphate (FPP), dolichol, and CoQ10. Decreases in isoprenoids FPP and GGPP result in decreased glucose transporter protein type 4 (GLUT4) glucose uptake. Reduced dolichol results in reduced insulin receptor membrane levels. CoQ10 is necessary to produce mitochondrial ATP to stimulate insulin secretion via calcium influx in pancreatic β-cells (Figure 3). Other possible mechanisms include decreased insulin signaling, decreased adipocyte differentiation via impaired peroxisome proliferator-activated receptor gamma (PPAR-γ) signaling, and decreased pancreatic β-cell insulin secretion [36]. Certain evidence suggests that lipophilic statins, which can passively diffuse into extrahepatic and hepatic cells, may worsen insulin sensitivity or inhibit insulin secretion. These statins include atorvastatin and simvastatin. Rosuvastatin and pravastatin, hydrophilic statins which require carrier-mediated uptake, may have a lesser effect on insulin sensitivity (Figure 4). It is important to note that, overall, the incidence of developing diabetes appears to be confined to those with impaired glucose tolerance and other diabetes risk factors [14].

Studies have been conducted investigating the incidence of NODM in patients undergoing statin therapy. A cohort study by Yoon et al. [10] conducted in Korea in 2016 investigated the risk of statin-induced NODM in real-world clinical practice. This study found that the development of NODM was significantly higher in the statin-treated group when compared to the non-statin-treated group. Additionally, this study found that male gender, baseline glucose level, hypertension, and thiazide use showed an increased risk of NODM, whereas angiotensin-converting enzyme (ACE) inhibitor or angiotensin II receptor blocker (ARB) showed a decreased risk [10].

A study by Choi et al. in 2018 investigated the effect of three popular statin drugs (2–4 mg daily pitavistatin, 10–20 mg daily atorvastatin, 5–10 mg daily rosuvastatin) on NODM in patients with acute myocardial infarction (AMI) [11]. Statin therapy is imperative for patients who have had an AMI for secondary prevention of cardiovascular events; therefore, it is important for research to be conducted in this specific patient population. The study found that the cumulative incidence of NODM was significantly lower in the pitavastatin group compared with the atorvastatin and rosuvastatin groups [11]. These data support the efficacy of pitavastatin as a potential preferred treatment for patients with a prior AMI who are at risk of developing diabetes. Overall, the literature demonstrates that hyperlipidemia treatment options need to be carefully considered for patients who have an established risk of developing diabetes.

### 2.4. Statin-Resistant Hyperlipidemia

While conducting a systematic review of selected publications, a patient population was identified that either could not tolerate a maximum statin dosing regimen or failed to achieve target lipid levels despite maximum lipid-lowering therapy [15,38,39,40]. Traditionally, patients with statin intolerance are those who cannot tolerate at least two statins, one being at a low dose, due to adverse side effects, most notably SAMS. A study by Rallidis et al. [38] further differentiated patients into subcategories, which included total and partial intolerance. Total intolerance is defined as an inability to tolerate any statin, while partial intolerance is the ability to tolerate a low-dose statin or intermittent dosing. Regardless of whether the intolerance is full or partial, nocebo effect is of little consequence. If the patient is unable or unwilling to take the medication, an alternative must be found. There is a separate population that fails to reach desired LDL-C levels, despite taking the maximal dose of statins [38]. While both groups are reported as having statin intolerance in the literature, there should be a distinction between the two groups. We and other groups have defined statin-resistant hyperlipidemia as LDL-C goals above target levels despite maximal statin dosing. While patients with statin intolerance or resistance may account for a modest percentage of those being treated for hypercholesterolemia, this presents a real challenge for providers helping patients achieve acceptable LDL-C levels.

Pharmacogenomic polymorphisms appear to have dramatic influences on not only baseline lipid levels and cardiovascular risk, but some also impact response to statins in terms of lipid reduction. The role of apoE is the regulation of lipid metabolism and involvement in the development of CVD is well-established. In apoE, the rs7412 mutation was strongly associated with LDL-C response [41]. *SLCO1B1* c.521T > C variant rs4149056 does not appear to impact statin response [17]. The effect of the EYS rs9342288, rs1337512, and rs3857532 on statin efficacy is poorly documented, likely due to the severity of side effects [16,42]. PCSK9 E670G polymorphism is an independent determinant of plasma LDL-C levels, severity of coronary atherosclerosis, and is associated with severity of intracranial atherosclerosis, so can be used as a predictor of large-vessel atherosclerosis [43,44]. Mutations in this gene have been associated with hypolipidemia through ‘‘loss-of-function’’ and hyperlipidemia through ‘‘gain-of-function’’ mechanisms. However, it does not appear to have any influence in response to either atorvastatin or pravastatin [45,46], and, similar to mutations in PCSK9 I474V, it has not been shown to be associated with lipid levels or CHD risk in healthy men [45]. PCSK9 Y142X and C679X are nonsense mutations associated with a lower LDL-C level and a 40 percent reduction in mean LDL cholesterol for black patients with the polymorphisms. These mutations are very rare in white subjects [47]. While a stop codon usually results in the production of a non-functional protein, consideration should be given to the fact that this is a beneficial mutation worthy of further study.

The PCSK9 R46L polymorphism is associated with a lower LDL-C level and significantly lower LDL-C in response to atorvastatin and pravastatin. The reduction in LDL-C and CHD has been shown and confirmed, and, in fact, reduction in risk of CHD was more than that predicted by reduction in LDL-C alone. Interestingly, R46L does not appear to alter response to rosuvastatin [45]. In one study, mean intima–media thickness was slightly but significantly lower among carriers of R46L mutations [47]. Rosuvastatin increased plasma concentration of PCSK9 in proportion to the magnitude of LDL-C reduction [48], which defies expected results of increased levels of PCSK9. This requires further study to determine the mechanism of this phenomenon.

## 3. Review of Non-Statin Treatment Options

Since the arrival of statins on the consumer market in 1987, they have remained an important cornerstone in the treatment of hypercholesterolemia and, ultimately, the prevention of ASCVD. The non-statin treatment that has become the focus of this literature review and treatment evaluation is PCSK-9 inhibitors. However, ezetimibe, red yeast rice, bempedoic acid and inclisiran should be mentioned as well for their antilipemic properties. The following section outlines major findings on the safety and efficacy of PCSK9 inhibitors with brief mentions of bempedoic acid, ezetimibe, and red yeast rice.

### 3.1. PCSK9 Inhibitors

Under normal physiological conditions, the PCSK-9 protein binds to LDL receptors (LDL-R) and promotes endocytosis and clearance of cholesterol by promoting LDL-R degradation (Figure 5). However, in the presence of a monoclonal antibody inhibitor, PCSK-9 does not bind LDL-R, resulting in more LDL-Rs present on the plasma membrane, thus allowing more clearance of lipoproteins and decreasing serum LDL-C [1,49]. Studies have shown that PCSK-9 inhibitors significantly lower LDL-C in patients as an adjunct to statin monotherapy. Additionally, there have been studies demonstrating that PCSK-9 inhibitors significantly lower the incidence of cardiac-related adverse events when added to statin therapy [50,51].

In a study conducted by Rallidis et al. [38], 75% of subjects had treatment-resistant hyperlipidemia, evidenced by failure to reach their target LDL-C levels with maximum tolerated lipid-lowering therapy (LLT), and the need for adjunct therapies. After placing patients on either evolocumab or alirocumab in addition to their existing statin therapy (at the healthcare providers discretion) for one year, 65.3% of subjects reached their LDL-C goals and the average reduction in LDL-C was 56 percent. This phenomenon was also described in the ODYSSEY EAST study (n = 615), which looked at patients in East Asia who had failed to reach their lipid targets despite maximal LLT [52]. The researchers placed selected patients on alirocumab and others on ezetimibe in conjunction with their statin. It was found that the alirocumab-treated group had an average 57 percent reduction in LDL-C from baseline while the ezetimibe-treated group had an average 22 percent reduction from baseline. As a primary efficacy endpoint, LDL-C was used to assess the efficacy of all lipid-lowering therapies. The systematic review of PCSK9 inhibitors showed markedly larger reductions in LDL-C in treatment groups receiving PCSK9 inhibitors when compared to other lipid-lowering therapies. Ezetimibe was also shown to lower LDL-C but not as effectively as PCSK9 inhibitors or bempedoic acid [53]. Moreover, ezetimibe monotherapy is limited in the literature.

In the ODYSSEY ALTERNATIVE study, patients with statin intolerance were randomized (2:2:1) to alirocumab (75 mg every two weeks subcutaneously, increased to 150 mg at week 12 plus oral placebo), ezetimibe (10 mg daily orally plus injectable placebo), or atorvastatin (20 mg daily orally plus injectable placebo) [53]. At the end of the treatment period, a significant change in LDL-C in both the ezetimibe (−14.6%) and alirocumab treatment groups was observed, with a notably larger reduction in LDL-C (−45%) observed in the alirocumab treatment group.

Similarly, all four GAUSS trials (2014–2019) compared the efficacy of evolocumab (PCSK9 inhibitor) and ezetimibe in patients with statin intolerance [54,55,56,57]. Across the four GAUSS studies reviewed, LDL-C reductions ranged from 41 to 56 percent in the PCSK9 inhibitor treatment groups compared to 15 to 39 percent LDL-C reductions seen in the ezetimibe treatment groups. It is important to mention that an average LDL-C reduction in 63 percent was reported in GAUSS 1 for the combined PCSK9 inhibitor/ezetimibe therapy (420 mg evolocumab/10 mg ezetimibe). Note that the percent changes in LDL-C are ranges reported across all GAUSS studies. The ODYSSEY EAST, MENDEL and MENDEL-2 trials and studies by Watts et al. [49], Ralidis et al. [38], and Cho et al. [15] also reported similar findings [39,40,52].

As a primary safety endpoint, adverse events were used to assess the safety of lipid-lowering therapies. Generally, adverse events were defined as those that developed, worsened, or became serious during treatment. In reference to the ODYSSEY ALTERNATIVE trial described above, adverse events were similar between both treatment groups (lowest in the alirocumab group by 4.0%) with the largest prevalence in the atorvastatin re-challenge group (11.1%) [53]. Furthermore, the GAUSS studies showed a 96 percent patient compliance rate and a significantly larger occurrence of adverse events in the ezetimibe group (23 percent) when compared to the evolocumab group (12 percent) [39,40,41,42]. However, there is a markedly larger disparity seen in studies where statin alternatives were directly compared to statin therapies. In the Watts et al.’s [49] study, adverse events were similar across treatment groups but with a greater frequency among the active intervention groups (70–86.4%) than for the placebo groups (52.4%). It is important to note that adverse events, particularly those associated with myalgia, were assessed by patient self-reporting. Lastly, while many of the studies reviewed provide promising short-term safety data, they were insufficiently long (<12 weeks in duration) to assess the long-term safety and tolerability of PCSK9 inhibitors. However, a recent study (FOURIER-OLE, 2022) that followed up on the 2017 FOURIER study, provided 8 years’ worth of follow-up data [58]. In the FOURIER study, patients with a history of peripheral artery disease at high risk of cardiovascular events were randomized to evolocumab versus placebo groups, while receiving background statin therapy [59]. The evolocumab group demonstrated significantly larger reductions in both LDL-C levels and CVEs. The FOURIER-OLE follow up study showed persistent LDL-C and CVE reductions for up to 8 years following the parent study. This demonstrates considerable promise for the long-term safety and efficacy of PCSK9 inhibitors [60,61,62,63,64,65,66,67].

### 3.2. Atheroprotective Properties of PCSK9 Inhibitors

Statin-based therapies remain the gold standard for treating hyperlipidemia, stabilizing atherosclerotic plaques, and lowering CVD risk in patients. Studies pairing a statin plus a PCSK9 inhibitor demonstrate greater antilipidemic effects. Yet, the question remains: do PCSK9 inhibitors reverse existing plaque and endothelial damage? Recently, studies have been conducted to explore the effects of non-statin therapies on plaque regression and plaque composition.

According to a meta-analysis [68] and two clinical studies [69,70], coronary plaque regression is significantly correlated with reductions in LDL-C and non-HDL-C levels. In both the Ota et al. [69] and Raber et al. [70] studies, intravascular ultrasonography (IVUS) showed that PCSK9 inhibitors significantly reduced coronary plaque when compared to placebo. This has further solidified the mainstay theory behind plaque regression which assumes a mechanism based solely on lipid reductions. That is, the greater the reduction in all lipoproteins, the more plaque regression will be observed. According to this theory, dual therapies (i.e., statin plus PCSK9 inhibitor), which have been shown to produce greater lipid reductions, should, therefore, contribute to greater plaque regression. Very few studies have explored this relationship to date. The data that are available are reviewed below.

In a pre-clinical study conducted by Pouwer et al. [71] investigating atheroprotective aspects of LLTs, female APOE3-Leiden CETP transgenic mice were treated with either a dual therapy (atorvastatin + alirocumab) or a monotherapy (atorvastatin). The dual therapy produced significantly greater reductions in plaque volume, composition, and morphology when compared to statin monotherapy alone [71]. In addition to the previously mentioned Ota et al. [69] study that assessed the atheroprotective effects of PCSK9 inhibitors on coronary plaque regression, the GLAGOV trial examined carotid plaque regression in patients receiving maximally tolerated LLT [72]. Both studies showed positive correlations between LDL-C reductions and plaque regression [69,70,71,72]. This further supports the “lower-the-better theory” for plaque regression and prevention of CVEs. Additionally, Xie et al. [73] claimed the PCSK9 protein itself may have atherogenic properties. According to their study, PCSK9 inhibitors contribute to plaque regression via lipid reduction effects as well as an LDL-independent mechanism involving inhibition of the PCSK9 protein [73]. The primary findings of the study showed a significant association between baseline PCSK9 levels and 10-year progression of carotid atherosclerosis.

Interestingly, a 2022 post-hoc analysis of ODYSSEY OUTCOMES conducted by White et al. [74] showed that the atheroprotective effects associated with PCSK9 inhibitors may vary across different patient populations. In the study, it was shown that while MACE was reduced in patients without a history of HF, that was not the case for patients with a history of HF. Unfortunately, the atherogenic mechanism associated with PCSK9 remains unclear. However, the role PCSK9 plays in the metabolism of VLDL as well as its effect on other atherogenic determinants (e.g., blood pressure, fasting glucose levels, CRP levels, and white blood cell count) are potential outcomes [73]. Lastly, the GLAGOV study showed that PCSK9 inhibitors, when combined with a maximally tolerated statin treatment plan, produced greater LDL-C reductions and increased atheroma regression compared to statin monotherapy [72]. In addition to total atheroma volume regression (TAV reduction of 5.5 mm^3^), PCSK9 inhibitors have also been shown to specifically reduce lipoprotein(a) by 20–30 percent. This is of particular importance given that lipoprotein(a) has been shown to be the primary cause of heart disease and is minimally affected by lifestyle changes or medications. Additionally, PCSK9 inhibitors were shown to reduce apolipoprotein B and TGs substantially more than the statin monotherapy group [50,59,75]. Similar claims have been made by the ongoing CARUSO study, which is currently investigating the effects of evolocumab on carotid plaque regression [76]. These findings illuminate a need to further investigate PCSK9 inhibitors as an adjunctive medication and possibly as a monotherapy in the treatment and prevention of hyperlipidemia and atherosclerosis.

### 3.3. Ezetimibe

Due to the incidence of statin intolerance, often the next drug of choice for lowering LDL-C is ezetimibe, a cholesterol absorption inhibitor [15]. However, therapeutic efficacy of ezetimibe is limited. In the original MENDEL study from 2012 and the follow up MENDEL-2 from 2014, average LDL-C reduction in patients taking ezetimibe monotherapy was 16.6% from baseline [39]. Patients who fail lifestyle modification and qualify for receiving statin treatment at the lowest level have serum LDL-C at 160 mg/dL or higher [2]. While this lipid value is improved from baseline, it is still not near the 20–50 percent reduction expected of patients on statin therapy [2]. It should be noted that in the MENDEL-2 study, subjects receiving evolocumab achieved an average reduction in LDL-C at –56% from baseline and over 70 percent of the treatment group had an LDL-C < 70 mg/dL at the end of the 12-week study. Adverse events leading to discontinuation of the study occurred in 3.9% of the placebo group, 3.2% of the ezetimibe group, and 2.3% of the evolocumab group. These data were deemed statically insignificant [39]. Discontinuation due to adverse events was lower in the Rallidis et al. [38] study (1.4%) and was 0% after two years in the OSLER studies [15]. Lastly, the IMPROVE-IT trial showed greater LDL-C reductions and improved cardiovascular outcomes in the dual therapy group consisting of atorvastatin and ezetimibe when compared to the monotherapy group consisting of atorvastatin alone [77]. This further demonstrates the benefits of adding ezetimibe or other LLTs to statin monotherapy regimens.

### 3.4. Bempedoic Acid

Bempedoic acid is a novel drug that is activated in the liver and inhibits adenosine triphosphate-citrate lyase (ACL), an enzyme involved in the cholesterol biosynthesis pathway (Figure 2) [78]. By inhibiting ACL, which acts upstream of the rate-limiting step for the cholesterol biosynthesis pathway, bempedoic acid decreases the amount of substrate available for cholesterol and fatty acid synthesis. This ultimately decreases liver cholesterol synthesis and decreases serum LDL-C levels by upregulating LDL receptors.

Bempedoic acid is a first-in-its-class medication that is taken once a day with a dosage range of 40 mg to 240 mg, with the most widely accepted dosage currently being the 180 mg pill. The half-life of bempedoic acid is 21 h and peak blood concentrations occur at 3.5 h after ingestion of the pill; it is also a prodrug that needs first-pass metabolism to be effective. Statins and bempedoic acid target the same pathway and are both cholesterol synthesis inhibitors, however, bempedoic acid targets ATP citrate lyase, which is upstream of HMG-CoA reductase, the target for statins (Figure 2) [78].

The CLEAR Wisdom study was a randomized, double-blinded, placebo-controlled trial that assessed the efficacy of bempedoic acid in patients already receiving maximally tolerated doses of statins for treatment of uncontrolled hypercholesterolemia. At the start of the study, all patients presenting with atherosclerotic cardiovascular disease and/or familial hypercholesterolemia were considered high risk for developing further cardiovascular disease. As indicated, the multi-society guidelines on management of blood cholesterol recommend maximally tolerated statin therapy in patients with atherosclerotic cardiovascular disease [79]. Bempedoic acid was chosen to supplement the maximum dose of statins to determine if the medication would be useful for patients with statin resistance. The patients in the trial that received 180 mg per day of oral bempedoic acid experienced an 18.1% decrease in LDL-C levels after 12 weeks of treatment [79]. This is a significant finding because these two drugs both target the cholesterol synthesis pathway and this study determined that they have an additive effect on LDL-C levels when given together. This information can be used to help patients that do not tolerate statins due to myalgia and other dose-dependent adverse events.

A three-phase study was conducted by Thompson et al. [80] to determine the effectiveness of combining bempedoic acid with ezetimibe. Phase 1 consisted of a 6-week screening period consisting of a 5-week single blind run-in period where all patients were screened for MACE. In phase 2, a 5-week washout of all lipid-regulating drugs/supplements was carried out and the patients refrained from continuing these drugs or supplements throughout the treatment period. Finally, in phase 3, the 12-week double-blind treatment period began. Patients were randomized 4:4:4:1:1 (n = 92:92:92:23:23) to the following treatment groups: 120 mg per day bempedoic acid; 180 mg per day bempedoic acid; 10 mg per day ezetimibe; 120 mg per day bempedoic acid plus 10 mg per day ezetimibe; 180 mg per day bempedoic acid plus 10 mg per day ezetimibe. Efficacy was measured as a percent change in LDL-C from baseline at the end of the 12-week treatment period for monotherapy treatments. Similarly, LDL-C levels, apolipoprotein B, and other lipid values were used to compare combined therapies with monotherapies. Safety was primarily measured using the number of treatment-emergent adverse events but also included hematology, serum chemistry, urinalysis, vital signs, EKG readings, and body weight. It was shown that both bempedoic acid monotherapy and ezetimibe combined therapy resulted in larger LDL-C reductions when compared to the ezetimibe monotherapy, with the greatest reductions seen with combined therapy group (*p* < 0.0001 vs. ezetimibe alone). There was a 43% LDL-C reduction seen in the 120 mg bempedoic acid group and a 48% LDL-C reduction in the 180 mg bempedoic acid group, with an average reduction in LDL-C of 30% [80]. It is important to note that LDL-C reductions were similar in patients with and without statin intolerance. Treatment-emergent adverse events in each bempedoic acid monotherapy group were similar to the ezetimibe monotherapy group. However, adverse events strongly attributed to the study drug were least common in the 120 mg bempedoic acid group and more common in the 180 mg bempedoic acid plus ezetimibe group. This study found that bempedoic acid is effective both in combination with ezetimibe and as a monotherapy, adding to the evidence that bempedoic acid may be a reasonable alternative to statin therapy.

Another three-phase, multicenter, randomized, double-blind, placebo-controlled study by Ballantyne et al. [81] was performed to evaluate the efficacy of bempedoic acid in the treatment of hyperlipidemia in patients that demonstrated statin intolerance or statin-resistant hyperlipidemia. The study design involved three phases: phase 1 of the study consisted of a one-week screening period where patients were selected based on inclusion criteria (statin intolerance, fasting LDL-C > 100 mg/dL). Phase 2 of the study consisted of a four-week run-in period where patients were given 10 mg daily ezetimibe. Phase 3 of the study consisted of a 12-week treatment period. During phase 3, patients were randomized 2:1 to a treatment group (low-dose statin therapy, 10 mg daily ezetimibe, plus 180 mg daily bempedoic acid) and a placebo group (low-dose statin therapy, 10 mg daily ezetimibe, NO bempedoic acid). It is important to note that all subjects in the study were receiving low-level background statin therapy or no statin therapy, as indicated by their treating physician before entering the study. The treatment group experienced a 28.5% decrease in LDL-C from 129 mg/dL at baseline to 96.2 mg/dL at the end of the 12-week treatment period. The placebo group experienced a slight increase from 123 mg/dL at baseline to 128.8 mg/dL at week 12. It is important to note that primary outcomes were first observed at week 4 and were maintained throughout the remaining weeks of the study. A subgroup analysis showed that the LDL-C-lowering effect of bempedoic acid was greater (34.7%) in the patients who were not receiving background lipid-lowering therapies (such as statins) other than the run-in treatment when compared with those receiving low-dose statin therapy. Lastly, bempedoic acid was also shown to improve other lipid and lipoprotein parameters (non-HDL-C, total cholesterol, and apoB). This is an important study mainly because it was able to determine that, while combining statins with bempedoic acid was effective at lowering LDL-C levels, bempedoic acid was more effective when combined with a non-cholesterol synthesis inhibitor, such as ezetimibe. This could make bempedoic acid a great alternative to statin therapy for patients with statin intolerance or resistance.

### 3.5. Red Yeast Rice

Increasing attention is being paid to nutraceuticals for prevention and management of hyperlipidemia. A 2020 study by Iskandar et al. [82] investigated the effects of a nutraceutical product containing red yeast rice extract on subjects who had moderate dyslipidemia. Red yeast rice has an identical structure to lovastatin and is an HMG-CoA inhibitor. At the conclusion of the study, there was a significant difference between total cholesterol and LDL levels of the nutraceutical-treated group when compared to the placebo-treated group. Previous clinical studies showed similar results.

The tolerability of this nutraceutical was also evaluated. There were no significant changes on renal and liver function parameters between baseline and study termination. No serious adverse events occurred during the study and, overall, the nutraceutical was tolerated by all subjects. The data from this study were supported by a recent meta-analysis which included more than 8500 subjects and showed that red yeast rice supplementation is safe and not associated with increased incidence of muscular adverse effects [83].

Red yeast rice products may provide an alternative cholesterol-lowering therapy for those patients who are not able to tolerate statins. Nutraceuticals may also be a useful complementary treatment for those patients who are only intolerant of high-dose statins but are able to tolerate lower doses. This study was conducted on subjects with mild dyslipidemia, so it is unknown whether nutraceuticals would benefit those with more severe hyperlipidemia.

### 3.6. Inclisiran

Another statin alternative, inclisiran, has completed stage III clinical trials and was approved for use in the US in late 2021 [84]. Inclisiran is a small interfering ribonucleic acid (siRNA) therapy delivered by subcutaneous injections every six months [85]. The drug is tagged with a triantennary N-acetylgalactosamine modification which targets asialoglycoprotein receptors expressed only on hepatocytes. Once in the hepatocyte via receptor-mediated endocytosis, the siRNA forms a complementary pairing with the RNA-induced silencing complex, which then ultimately inhibits translation of the PCSK9 mRNA into protein [86]. The ORION studies enrolled patients with statin-resistant hyperlipidemia as subjects and established a greater than 50 percent reduction in mean LDL-C levels with a dosing of one injection every six months. There was also a significant decrease in non-HDL cholesterol and apolipoprotein B. Adverse events occurred in similar frequency between the placebo and treatment groups [85]. While this drug is new and more data are needed on plaque regression, lipoprotein(a) and HDL levels, and even long-term data for anti-inclisiran antibodies, the idea that hyperlipidemia can be controlled with injections every six months is hopeful for increased patient compliance and better long-term outcomes, particularly for the population with statin intolerance or resistance.

## 4. Methods

### 4.1. Systematic Search Strategy

PubMed and Google Scholar were searched from September 2019 through November 2022 to gather pertinent information regarding treatment of hyperlipidemia. Search key words included PCSK9 inhibitors, statin intolerance, non-statin therapy, hyperlipidemia, and statin alternatives. A total of 40 studies (36 clinical and 4 preclinical) were identified as relevant to this project. Additionally, 9 meta-analyses consisting of 48 studies from 2012 to 2018 were analyzed. While searching for drugs to include in the meta-analysis, several drugs in clinical trials were found. However, only those available for treatment, the PCSK9 inhibitors alirocumab and evolocumab, were chosen. It was determined there was a paucity of information regarding average decrease in LDL-C, increase in HDL-C, and incidence of non-cardiac adverse events. During the selection process, after papers were evaluated for similarities between the treatment and control group and statistically significant results, the following inclusion criteria were used: (1) phase 1, 2 or 3 human randomized control clinical trials; (2) PCSK9 inhibitors were compared against either a placebo or non-statin treatment regimen; (3) data were recorded for non-cardiac-related adverse events, LDL reduction, and HDL increase (4) all studies lasted longer than 12 weeks.

### 4.2. Systematic Review of 9 Meta-Analyses

After completing a preliminary review of our 36 clinical and 4 preclinical studies, an additional review of 9 meta-analyses was conducted. These meta-analyses included any study involving treating patients with hypercholesterolemia with PCSK9 inhibitors. Knowing the focus was specifically on treatment of patients with statin intolerance or statin-resistant hyperlipidemia, it was necessary to identify if there was a gap in the general knowledge regarding treatment of these patients with PCSK9 inhibitors. The most common topics investigated by the meta-analyses were all-cause mortality, major adverse cardiac events (MACE), myocardial infarction (MI), and neurocognitive adverse effects (including strokes). All these categories were reported in an odds ratio (OR) comparing PCSK9 inhibitor adjuvant therapy to statins. However, there seemed to be a paucity in the meta-analyses regarding the specific decrease in LDL-C, increase in HDL-C and incidence of non-cardiac-related adverse events. The most recent paper published that was included in these meta-analyses was from 2018. This proposed meta-analysis would include data up to 2021, thus adding 3–5 years of data regarding non-cardiac-related adverse events, and up to 5–6 years of data regarding serum lipid levels. Thus, a meta-analysis investigating PCSK9 inhibitor monotherapy in patients with statin intolerance or statin-resistant hyperlipidemia was determined.

### 4.3. Statistical Analysis

All chosen studies were pooled together for analysis and the OR for the endpoints was determined. Comparison of 95% confidence intervals was used to establish statistical significance for all comparisons between ORs. The ORs analyzed were for incidence of myalgia and MI. The software used for pooling, analyzing, and creating graphical representations of the data was ProMeta 3 (IDoStatistics). Funnel and forest plots were used to visualize the data.

Meta-analyses were performed on eight studies looking at incidence of myalgia and five studies looking at incidence of myocardial infarction. *p* < 0.05 was considered significant.

## 5. Results

### 5.1. Results of Meta-Analysis: Myalgia

After sorting through 36 clinical trials as well as the 48 included in the 9 meta-analyses, 13 studies were eligible for this meta-analysis. The primary endpoint in this meta-analysis was to evaluate the incidence of muscular adverse events, namely myalgia. Through a systematic search, eight studies qualified for inclusion, which yielded a total sample of 1724 subjects.

When comparing patients receiving subcutaneous PCSK9 inhibitors to control groups receiving placebo with or without ezetimibe or placebo plus maximally tolerated statin, the OR was 0.91. However, this combined result was not statistically significant. PCSK9 inhibitors showed no discernable difference in events of treatment-emergent myalgia when compared to control groups, suggesting that there is no increased incidence of myalgia induced by PCSK9 inhibitors (Figure 6 and Figure 7).

### 5.2. Results of Meta-Analysis: Myocardial Infarction

If PCSK9 inhibitors are to be effective treatments for patients with statin intolerance, they must also decrease LDL-C levels to ultimately reduce the incidence of cardiovascular events. This meta-analysis, therefore, used MI as a measure of PCSK9 inhibitor efficacy. Five studies were identified that fit the inclusion criteria, had data for MI, and consisted of 1362 subjects. When subjects receiving subcutaneous PCSK9 inhibitor treatment were compared to the placebo plus ezetimibe group, the OR for MI was 0.68. However, the result is not statistically significant. This suggests that PCSK9 inhibitors do not cause additional harm, expressed as cardiac events, to patients taking these drugs. The difference between cardiovascular events of patients on PCSK9 inhibitors and those in the placebo group was statistically insignificant (Figure 8 and Figure 9).

## 6. Discussion

### 6.1. Clinical Significance

Cardiovascular disease is the leading cause of death for adults in the United States, according to the Centers for Disease Control and Prevention [1]. Patients with hyperlipidemia are at roughly twice the risk of developing cardiovascular disease compared to those with normal cholesterol levels. Currently, seven FDA-approved statins have become the cornerstone of hyperlipidemia treatments. However, there are limitations to statins which include treatment resistance and intolerance due to adverse events. For these patients experiencing resistance or intolerance, it is important that there are alternatives to statins to prevent adverse events. With respect to LDL-C, the treatments being explored are as follows: statin monotherapy, non-statin monotherapy, and combined treatment plans.

Overall, the analyzed data show that PCSK9 inhibitors are generally well tolerated and are successful at lowering LDL-C levels typically between 50–60 percent as an adjunctive therapy [38]. The meta-analysis demonstrates that incidence of myalgia is not statistically different between PCSK9 inhibitor-treated groups and control groups and, therefore, it is not likely that PCSK9 inhibitors will induce myalgia any more than conventional treatment. This is promising for the treatment of patients who have statin intolerance.

By lowering LDL-C levels, the risk of developing cardiovascular events will also be reduced. The meta-analysis data demonstrate that PCSK9is show no statistical inferiority to statins in preventing cardiovascular events (measured by incidence of MI). However, the GLAGOV clinical trial evaluated the effects of PCSK9 inhibition using evolocumab on progression of coronary atherosclerosis [72]. This study had a medium-term follow-up period (78 weeks), a large sample population (n = 968), and statistically significant outcomes. The percent atheroma volume (PAV) increased 0.05% with placebo and decreased 0.95% (*p* < 0.001) with evolocumab [60]. Overall, evolocumab induced plaque regression in a greater percentage of patients than placebo. These conclusions are promising for physicians attempting to treat patients with statin intolerance. It is important to note that while this review focused on the populations with statin intolerance, these PCSK9 inhibitors can also be combined with statins to further reduce LDL-C. It is also important to note the significance of emphasizing these medications as adjunct therapies to proper diet and exercise when prescribing to patients. While PCSK9 inhibitors show significant potential in their ability to treat hyperlipidemia in a safe and efficacious manner, it should be mentioned that the monthly cost of these drugs is currently prohibitive [87]. There are additional barriers to accessing PCSK9 inhibitors besides cost. Investigating prescribing practices in ten countries in Europe, Asia, and the continent of Australia, access barriers were identified, including reservation of the drug for only high- or very high-risk cohorts and restriction of prescribing authority to specialists only (as opposed to general practitioners) [88]. In the United States, access barriers included the need for proper documentation for insurance companies to cover the medications, administrative burden of submitting/resubmitting paperwork, and the appeals process [89].

### 6.2. Limitations and Remarks

While the data generated by ProMeta 3 showed PCKS9is as comparable to statin therapy regarding adverse events, it is important to note pertinent limitations. These limitations should be remedied in future studies and reviews to solidify the analysis of the safety and efficacy of PCSK9 inhibitors as a lipid-lowering therapy in a patient population with statin intolerance. The five major limitations that were identified are as follows: limited number of parameters investigated, population size and significance, population homogeneity, reliability of current endpoints, and lack of long-term study parameters.

Only two side effects were investigated within the scope of this meta-analysis: incidence of myalgia and incidence of MI. It would be prudent to investigate various other side effects, including but not limited to all TEAE, other MACE, incidence of type 2 diabetes, and mortality. In addition to the limited number of side effects, it is also important to bring attention to the duration of the studies being reviewed and the effect of the study duration having no long-term parameters such as incidence of MI or mortality. Due to the relatively short study duration (approximately 1–2 years), the long-term harms and benefits remain unexplored. Therefore, future meta-analyses investigating the role of PCSK9 inhibitors as lipid-lowering monotherapy in populations with statin intolerance should include all relevant parameters and be conscious of the importance of reviewing long-term studies. It is also important to mention that some of the studies considered as part of the meta-analysis were designed specifically for outcomes directly related to one parameter but did not report data for the second parameter. To maintain homogeneity, these studies were removed from consideration since they did not fit the inclusion criteria.

Secondly, the overall population size of 1724 subjects is relatively small when compared to other meta-analyses. This is due in part to the distinctiveness of population parameters (i.e., statin intolerance/resistance) and the lack of studies with a focus on statin intolerance and resistance. By defining a unique subset of hyperlipidemia patients as the target of the meta-analysis, it became increasingly difficult to obtain enough data without sacrificing the homogeneity of the patient population. Therefore, future meta-analyses might benefit from a more broadly defined patient population to include a larger number of patients.

Thirdly, treatment homogeneity became difficult to maintain with increasing specificity of the patient population. With respect to the experimental population (i.e., patients receiving PCSK9 inhibitors), the patients were not all placed on a uniform treatment program. The type, dose, and dose frequency of PCSK9 inhibitors varied across studies. With respect to the control population, the goal was to define a control population that was receiving a placebo drug without any other lipid-lowering therapies. Unfortunately, it was difficult to define such a control without removing several studies from consideration. As a result, the control population was redefined to include patients receiving placebo, ezetimibe plus placebo, or ezetimibe plus maximally tolerated statins. In doing so, the overall patient population was not compromised. However, future meta-analyses might benefit from including more patients receiving low-dose statin therapies within the control population to improve the power of the results.

As a final remark, the endpoints and parameters assessed in many studies are not without their own limitations. The specific parameters in question include LDL-C as a primary endpoint and genetic factors such as pharmacogenetics. The reliability of LDL-C as a short-term endpoint for a long-term disease such as cardiovascular disease lacks utility for clinicians when compared to other endpoints such as serum lipoprotein(a) and apolipoprotein B, or plaque regression endpoints such as atheroma volume reductions. Lastly, for the sake of completeness, assessing pharmacogenetic variabilities would further establish the safety and efficacy of PCSK9 inhibitors. In continuation of the goals of this paper, it is advised that further attention be given to these endpoints and parameters to help researchers and clinicians solidify the comprehensive role of PCSK9 inhibitors in populations with statin intolerance or statin-resistant hyperlipidemia.

## 7. Conclusions

Of the many statin-alternative drugs on the market or in clinical trials, a systematic review of the literature elucidated the safety and efficacy of PCSK9 inhibitors both as a statin-adjunctive therapy and as a monotherapy. Our meta-analysis demonstrates there is no significant difference between the incidence of myalgia in patients treated with PCSK9 inhibitors compared to control patients. Our meta-analysis also demonstrates that the difference between MIs of patients on PCSK9 inhibitors and those in a placebo group was statistically insignificant. PSCK9 inhibitors show promise in patients with statin intolerance and resistance, however, more research with larger and more homogenous populations is needed to truly elucidate their long-term efficacy. While PCSK9 inhibitors were the primary focus of this review, it is important to give recognition to the previously mentioned statin-alternative medications (ezetimibe, bempedoic acid, red yeast rice and inclisiran) as worthy candidates for investigation in future studies. With the prevalence of patients with intolerance or resistance to statin medications, clinicians require statin alternatives for their patients. Fortunately, the future of LLTs, both as a mono or dual therapy, shows promise for physicians and patients alike, given the growing number of options becoming available.

## Figures and Tables

**Figure 1 ijerph-19-16899-f001:**
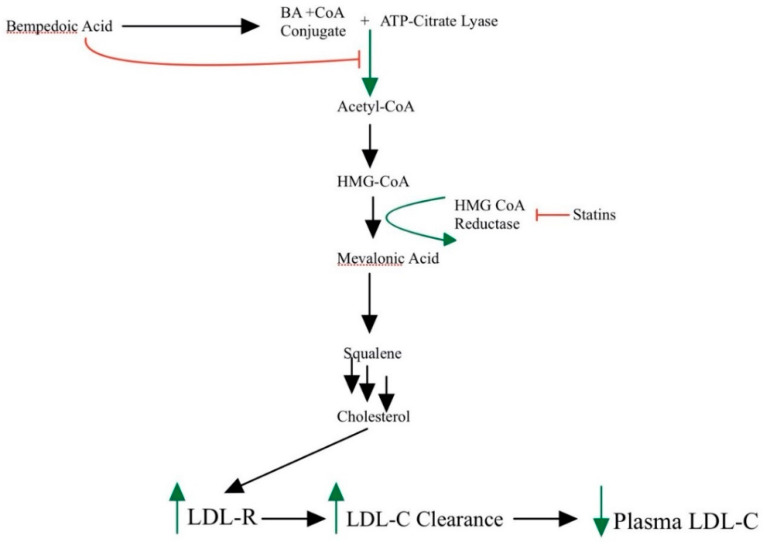
Inhibition of cholesterol synthesis pathway by bempedoic acid and statins. Bempedoic acid inhibits ACL upstream of HMG CoA reductase (**top** of figure). Statins inhibit HMG CoA reductase downstream of ACL (**center** of figure). BA and statins promote the upregulation of LDL receptors thereby increasing LDL-C clearance and reducing plasma LDL-C (**bottom** of figure). ACL, ATP citrate lyase; BAA, bempedoic acid; HMG-CoA, β-hydroxy β-methylglutaryl-CoA; LDL-C, low-density lipoprotein cholesterol; LDL-R, low-density lipoprotein receptor.

**Figure 2 ijerph-19-16899-f002:**
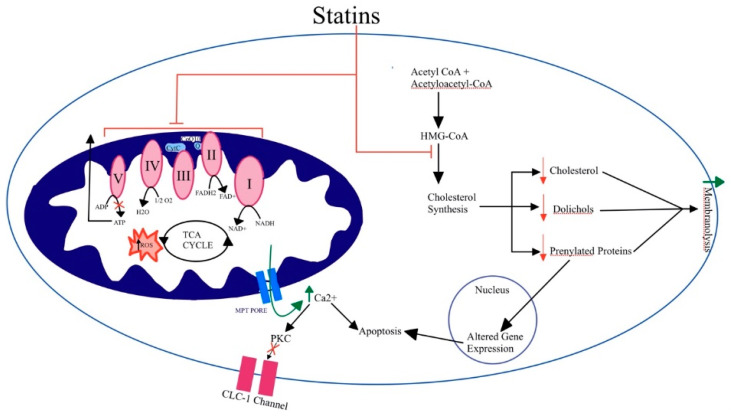
Proposed mechanism for statin-induced myopathy. Impaired calcium signaling increases mitochondrial caspase activity, and apoptosis results in CLC-1 channel inhibition and hyperexcitation of cell membrane (**left** of figure). Inhibition of the cholesterol synthesis pathway alters membrane composition, promoting membranolysis (**right** of figure). ATP, adenosine triphosphate; CLC-1, chloride channel 1; FAD, flavin adenine dinucleotide; HMG-CoA, β-hydroxy β-methylglutaryl-CoA; MPT pore, mitochondrial permeability transition pore; PKC, protein kinase C; TCA, tricarboxylic acid.

**Figure 3 ijerph-19-16899-f003:**
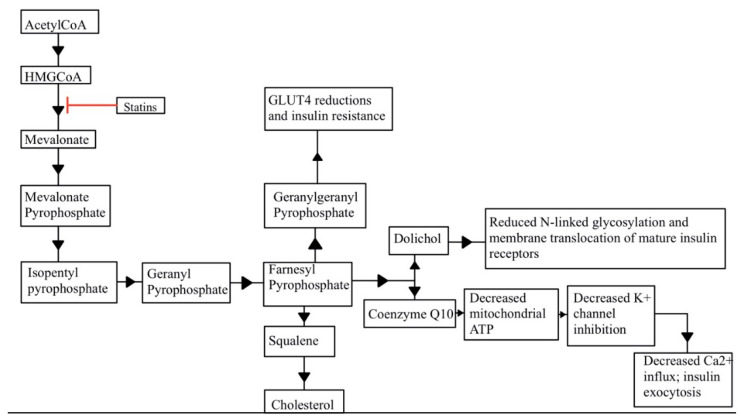
Proposed pathways for statin-induced type 2 diabetes mellitus via inhibition of the cholesterol synthesis pathway. Reductions in GLUT 4 receptors results in decreased insulin sensitivity (**center** of figure). Reduced translocation of insulin receptors results in decreased insulin sensitivity (**center right** of figure). Reduced calcium influx results in decreased insulin secretion (**bottom right**). ATP, adenosine triphosphate; GLUT 4, glucose transporter type 4; HMG-CoA, β-hydroxy β-methylglutaryl-CoA.

**Figure 4 ijerph-19-16899-f004:**
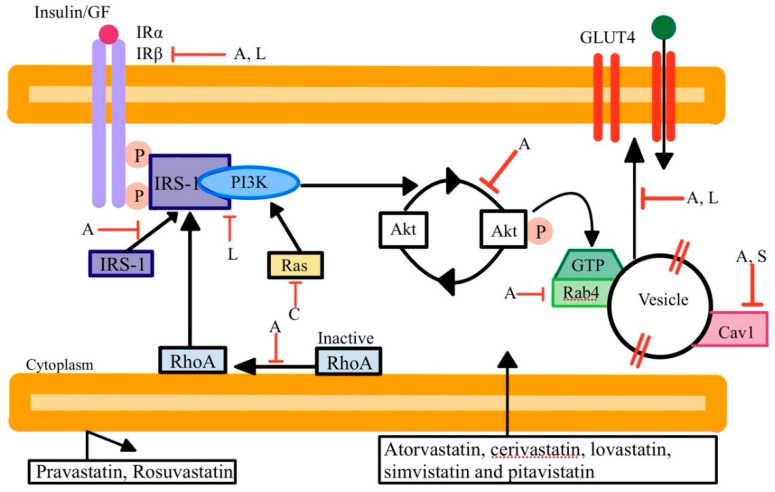
Proposed mechanisms of statin-induced type 2 diabetes mellitus. Hydrophobic statins (pravastatin, rosuvastatin) require active transport to enter the cell (**left** of figure). Lipophilic statins (atorvastatin, cerivastatin, lovastatin, simvastatin, pitavistatin) can freely diffuse across the cell membrane (**right** of figure). The first letter of each statin is listed where it acts in the insulin secretion pathway. Akt, protein kinase B; A, Atorvastatin; Cav1, caveolin 1; C, Cerivastatin; GF, growth factor; GLUT 4, glucose transporter type 4; IR, insulin receptor; IRS-1, insulin receptor substrate 1; L, Lovastatin; P, phosphate; PI3K, phosphoinositide 3 kinase; Rab4, ras-associated binding protein 4; Ras, rat sarcoma; S, Simvastatin.

**Figure 5 ijerph-19-16899-f005:**
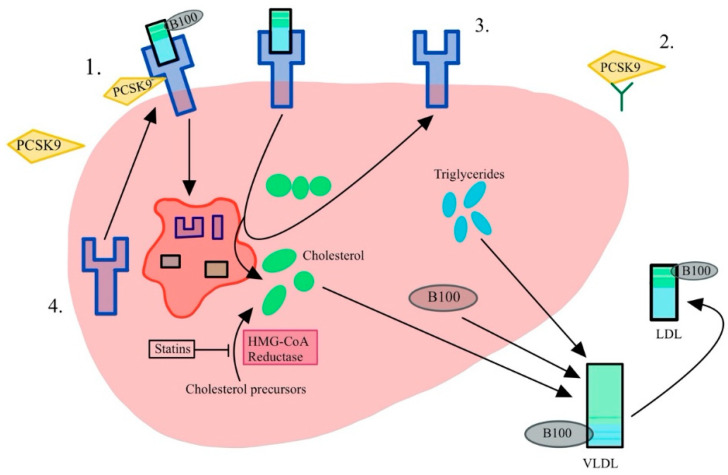
The mechanism of action of PCSK9 inhibition in the hepatocyte. (1) PCSK9 marks the LDL receptor for endocytosis and lysosomal degradation. (2) An anti-PCSK9 antibody prevents PCSK9 binding to the LDL-R. (3) The free LDL-R clears more LDL particles. (4) A decrease in intracellular cholesterol due to statins will upregulate LDL-Rs and subsequent clearance. HMG-CoA, β-hydroxy β-methylglutaryl-CoA; LDL, low-density lipoprotein; PCSK9, proprotein convertase subtilisin/kexin type 9; VLDL, very low-density lipoprotein.

**Figure 6 ijerph-19-16899-f006:**
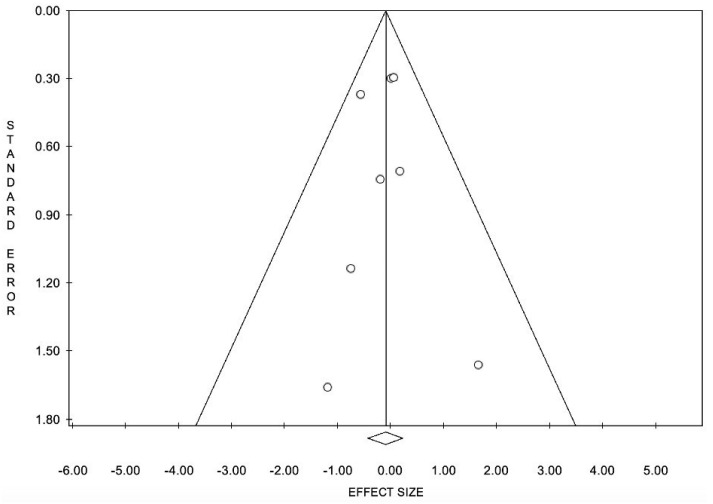
Funnel plot for meta-analysis of incidence of myalgia. Funnel depicts no publication bias for meta-analysis of myalgia incidence in anti-PCSK9 group vs. control group.

**Figure 7 ijerph-19-16899-f007:**
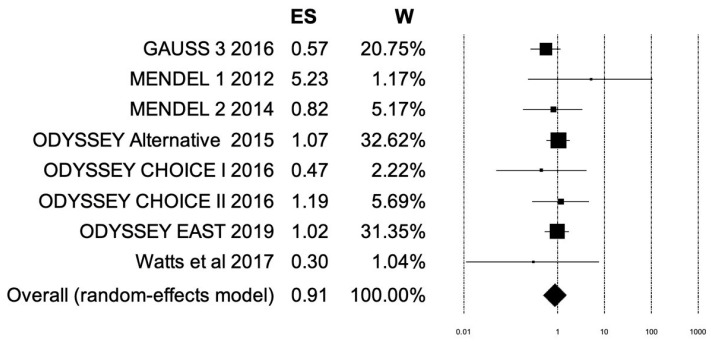
Forest plot for meta-analysis of myalgia incidence. Forest plot depicting effect size and weight for each of the studies included in the meta-analysis of myalgia incidence in anti-PCSK9 group vs. placebo group. Graphically, the effect size is represented as the center of each box while the box size represents the weight of each study and the whiskers represent the 95% confidence interval for each study. Center line (ES = 1) is the line of no effect where intervention (PCSK9 inhibitors or placebo) has no effect on outcome (no incidence of myalgia). The overall ES of 0.91 favors PCSK9 inhibitors. ES, effect size; W, weight [49].

**Figure 8 ijerph-19-16899-f008:**
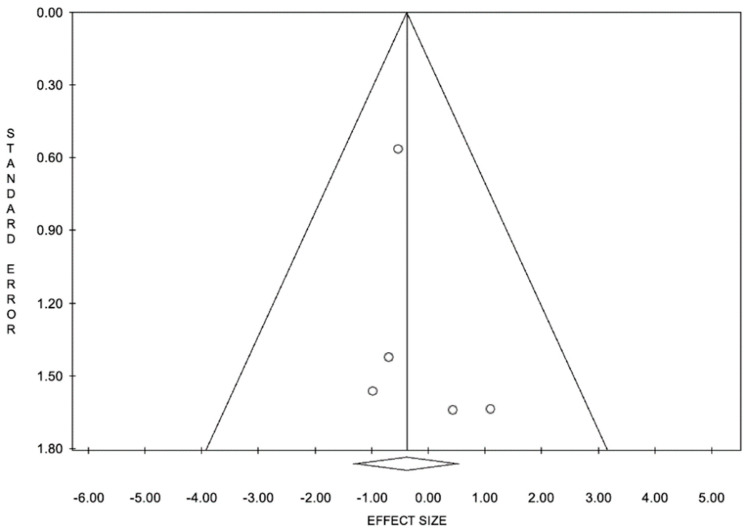
Funnel plot for meta-analysis of incidence of myocardial infarction. Funnel depicts no publication bias for meta-analysis of incidence of myocardial infarction in anti-PCSK9 group vs. control group.

**Figure 9 ijerph-19-16899-f009:**
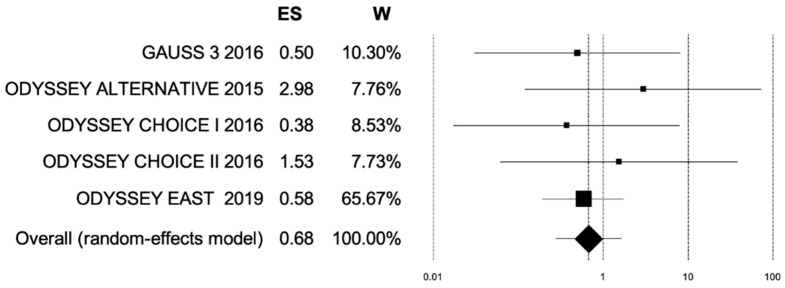
Forest plot for metanalysis of incidence of myocardial infarction. Forest plot depicting effect size and weight for each of the studies included in the meta-analysis of myocardial infarction incidence in anti-PCSK9 group vs. placebo group. Graphically, the effect size is represented as the center of each box while the box size represents the weight of each study and the whiskers represent the 95% confidence interval for each study. Center line (ES = 1) is the line of no effect where intervention (PCSK9 inhibitors or placebo) has no effect on outcome (no incidence of myocardial infarction). The overall ES of 0.68 favors PCSK9 inhibitors. ES, effect size; W, weight.

## Data Availability

Data are available upon request to the corresponding author.

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
