# Peer review of "Is a PCSK9 Inhibitor Right for Your Patient? A Review of Treatment Data for Individualized Therapy"

_ijerph, 2022, doi:10.3390/ijerph192416899_

Round 1

Reviewer 1 Report

In abstract specify that you are referring to population of adults where you quote >50% ar affected by hyperlipidemia - that figure seems remarkably high. How does this compare with other countries? The first paragraph of this article focuses on 'American society', is the article tailored for American readership or an international audience?

Vitamin D section probably too long with few paragraphs not directly relevant/not pertinent to convey message in this article.

With regards to HF classification, both ischemic and non-ischemic HF can be classified into HFrEF and HFpEF, but also worth mentioning HFmrEF. Suggest remove 'diastolic' in front of diastolic HFpEF.

Was systematic review prospectively registered (e.g. PROSPERO)? Systematic review search date ended in June 2021 (which was >12 months ago), which seems like a long time interval to this submission.

With regards to "low LDL-C levels may be correlated with an increased severity of HF", suggest briefly discuss "cholesterol paradox" [also discussed in reference 5].

I disagree with sentence 'statin therapy may be correlated with an increased severity of heart failure... an increased mortality in heart failure patients treated with statins [I don't see evidence of this from the cited reference 5].

Suggest include IMPROVE-IT data on ezetimibe. https://www.nejm.org/doi/full/10.1056/nejmoa1410489 

Apart from cost, what are barriers in more widely implementing the use of PCSK9 inhibitor?

Personally, I think 'summary statistics' Figures are not required if forest plots are provided.

Author Response

Comments raised by Reviewer 1.

 We would like to thank you for your time and consideration in reviewing our manuscript.  We appreciate the recommendations, which improved the quality of our manuscript.

1-In abstract specify that you are referring to population of adults where you quote >50% are affected by hyperlipidemia - that figure seems remarkably high. How does this compare with other countries? The first paragraph of this article focuses on 'American society', is the article tailored for American readership or an international audience?

We thank you for bringing this to our attention. While the first paragraph does focus on the prevalence of hyperlipidemia in American society, the content of the paper is applicable to an international audience as hyperlipidemia is prevalent internationally. We have changed this section.

Revised (Abstract):

In the United States, a significant amount of the population is affected by hyperlipidemia, which is associated with increased levels of serum low density lipoprotein (LDL-C) and risk for cardiovascular disease. 

2-Vitamin D section probably too long with few paragraphs not directly relevant/not pertinent to convey message in this article.

We discussed the role of vitamin D on myopathy since aberrations in calcium metabolism and vitamin D are linked to myalgia. We agree that this section is too long, and we have shortened it to address this relationship.

Revised section (page 5 of clean version):

Vitamin D is an important modulator of calcium metabolism.  It is common knowledge since the 1970’s that vitamin D deficiency even in the absence of statin use can predispose patients to myalgias and poor muscle function.  Serum levels below 30 nmol/L (12 ng/mL) are associated with severely impaired muscle function and decreased muscle strength [24,25]. The prevalence of vitamin D deficiency in statin-induced myalgias has been reported.  Statin-induced myalgias were resolved by correcting serum vitamin D (expressed as 1,25(OH)2D) levels [26-28].  These findings were later confirmed in a study involving patients with intolerance to 2 or more statins because of myalgia, myositis, myopathy, or myonecrosis were found to have low (<32 ng/mL) serum 1,25(OH)2D levels.  After correction to 50-80 ng/mL, statins were reintroduced in these patients and titrated to bring LDL-C to below the Adult Treatment Panel (ATP) III guidelines. After a period of 24 months, 78 of the 146 patients remained myalgia-free [29]. Further studies and meta-analyses continue to suggest similar associations [30].

3-With regards to HF classification, both ischemic and non-ischemic HF can be classified into HFrEF and HFpEF, but also worth mentioning HFmrEF. Suggest remove 'diastolic' in front of diastolic HFpEF.

We agree and have made the appropriate corrections based on the latest definitions. We also added the Consensus Statement on the classification of heart failure by Bozkurt et al. published in the Journal of Cardiac Failure (2021).

Revised statement (page 5 of clean version):

Non-ischemic HF can also be further classified based on ejection fraction (EF) as preserved EF (HFpEF), reduced EF (HFrEF), mildly reduced (HFmrEF), and improved EF (HFimpEF).

4-Was systematic review prospectively registered (e.g. PROSPERO)? Systematic review search date ended in June 2021 (which was >12 months ago), which seems like a long time interval to this submission.

We appreciate your comment and agree that there is a gap between completion and submission of our review. We have surveyed PubMed to include the latest papers. The review now includes all relevant studies up until November 2022. The studies added to this review are listed below. We have added a total of 8 studies. All related literature not reviewed in this paper either has a focus that is out of the scope of this paper or reiterates information already included in this paper.

Additional papers included in the revised review are listed below and cited as references no. 5, 6, 58, 70, 74, 77, 88, 89.

Ref. no. 5

Vallianou NG, et al. Statins and cancer. Anticancer Agents Med Chem. 2014; 14(5):706-12. doi: 10.2174/1871520613666131129105035.

Ref. no 6.

Chang HL, et al. Simvastatin induced HCT116 colorectal cancer cell apoptosis through p38MAPK-p53-survivin signaling cascade. Biochim Biophys Acta. 2013;1830(8):4053-64. doi: 10.1016/j.bbagen.2013.04.011.

Ref. no. 58

O’Donoghue ML, et al. Long-term evolocumab in patients with established atherosclerotic cardiovascular disease. Circ. 2022;146:1109–1119. doi: 10.1161/CIRCULATIONAHA.122.061620

Ref. no. 70

Raber L, et al. Effect of alirocumab added to high-intensity statin therapy on coronary atherosclerosis in patients with acute myocardial infarction: the PACMAN-AMI randomized clinical trial. JAMA. 2022;327(18):1771-1781. doi: 10.1001/jama.2022.5218

Ref. no. 74

White HD, et al. Alirocumab after acute coronary syndrome in patients with a history of heart failure. Eur Heart J. 2022;43(16):1554-1565. doi: 10.1093/eurheartj/ehab804.

Ref. no. 77

Cannon CP, et al. Ezetimibe added to statin therapy after acute coronary syndromes. N Engl J Med. 2015;372:2387-2397. doi: 10.1056/NEJMoa1410489

Ref. no. 88

Apostolou EA, et al. Potential barriers in lipid-lowering treatment with PCSK9 inhibitors from a healthcare perspective. Comparative evidence from ten countries. European Heart Journal.  2022;43:2348. doi.org/10.1093/eurheartj/ehac544.2348.

Ref. no. 89

Cohen JD, et al. Barriers to PCSK9 inhibitor prescriptions for patients with high cardiovascular risk: Results of a healthcare provider survey conducted by the National Lipid Association. J Clin Lipidol. 2017: 11(4):891-900. doi: 10.1016/j.jacl.2017.04.120.

5-With regards to "low LDL-C levels may be correlated with an increased severity of HF", (page 6) suggest briefly discuss "cholesterol paradox" [also discussed in reference 5].

See addition (page 6 of clean copy):

Among patients with systolic heart failure, lipid-lowering therapy with rosuvastatin was not associated with a reduction in the primary endpoint of CV death, MI, or stroke at a median follow-up of 32.8 months compared with placebo. Despite effectively reducing LDL, triglycerides, and CRP with rosuvastatin, there was no impact on clinical events, other than CV hospitalizations. These findings were observed despite the high levels of prior coronary disease in the population. The authors noted that low total cholesterol levels in this population have actually been associated with worse outcomes in prior studies. While there was no significant benefit on clinical events, except reduction in hospitalization for CV causes and heart failure, there was also no evidence of harm associated with the reduction in LDL levels in the present study of older patients with relatively severe systolic dysfunction [37].

Kjeksus J et al. Rosuvastatin in older patients with systolic heart failure. NEJM, 2007, 357:2248-61.

6-I disagree with sentence 'statin therapy may be correlated with an increased severity of heart failure... an increased mortality in heart failure patients treated with statins [I don't see evidence of this from the cited reference 5].

We appreciate your comment. We removed references to the cholesterol paradox and reemphasized current guidelines.

However, once ischemia has occurred and a patient has developed some degree of HF, there is debate on the role of statins use in HF. While statins do decrease the odds for developing ischemic heart disease by lowering the rates of atherosclerosis, current guidelines do not advocate for initiating new HF patients on a statin, as the data does not show a significant decrease in mortality [8].

7-Suggest include IMPROVE-IT data on ezetimibe. https://www.nejm.org/doi/full/10.1056/nejmoa1410489 

We appreciate the recommendation and have included the IMPROVE-IT data in the manuscript (Cannon CP et al (ref 77) on the last page of references).

Lastly, the IMPROVE-IT trial showed greater LDL-C reductions and improved cardiovascular outcomes in the dual therapy group consisting of atorvastatin and ezetimibe when compared to the monotherapy group consisting of atorvastatin alone [77]. This further demonstrates the benefits of adding ezetimibe or other LLTs to statin monotherapy regimens.

8-Apart from cost, what are barriers in more widely implementing the use of PCSK9 inhibitor?

Personally, I think 'summary statistics' Figures are not required if forest plots are provided.

Thank you and we wish to expand on this.

There are additional barriers to accessing PCSK9 inhibitors besides cost. Investigating prescribing practices in ten countries in Europe, Asia and the continent of Australia, access barriers were identified, including reservation of the drug for only high or very high-risk cohorts and restriction of prescribing authority to specialists only (as opposed to general practitioners) [88]. In the United States, access barriers included the need for proper documentation for insurance companies to cover the medications administrative burden of submitting/resubmitting paperwork and the appeals process [89].

We have removed the summary statistics (figures 6 and 7).

Reviewer 2 Report

The manuscript revises in depth the pros and cons of the different strategies to manage LDL-C, with special attention on the introduction of PCSK9 inhibitors in this scheme. The review is organized in a sequential mode and myalgia associated with statins consumption appears to be the main determinant of the choice of hypercholesterolemic drugs.

If possible, I suggest being more precise on the dosage of statins. This is very dependent on the cardiovascular health of the patients’. As an additional aspect in the description of the side effects of statins, it is important to mention the actions independent of the control of cholesterol synthesis, as it involves acylation reactions relevant to cancer prevention. Also, there is a cross-talk between PCSK9 transcriptional regulation by statins that is relevant to be mentioned.

Apart from this, it is well-known that PCSK9 synergizes with pro-inflammatory pathways that, in the end, contribute to the generation of ROS and oxidized forms of LDL-C. In addition, the authors should consider the contribution of the different polymorphisms in the human PCSK9 and the gain/loss of activity as an unmet determinant in the future development of PCSK9 targeting molecules.

Finally, a comment on combined therapy with low doses of the different cholesterol-reducing drugs/approaches deserves attention, at least from an academic and medical point of view.

Author Response

Comments raised by Reviewer 2.

We would like to thank you for your time in reviewing our manuscript as well as your recommendations, which improved the quality of our manuscript.

The manuscript revises in depth the pros and cons of the different strategies to manage LDL-C, with special attention on the introduction of PCSK9 inhibitors in this scheme. The review is organized in a sequential mode and myalgia associated with statins consumption appears to be the main determinant of the choice of hypercholesterolemic drugs.

Thank you.

1-If possible, I suggest being more precise on the dosage of statins. This is very dependent on the cardiovascular health of the patients. As an additional aspect in the description of the side effects of statins, it is important to mention the actions independent of the control of cholesterol synthesis, as it involves acylation reactions relevant to cancer prevention. Also, there is a cross-talk between PCSK9 transcriptional regulation by statins that is relevant to be mentioned.

Part 1. Response regarding the precise dosage:

Not all references provided the dosages. In some studies, dosages were not provided based on the nature of the studies (reviews, longitudinal follow-up studies, etc.).  Where the specific dosages are cited (and maintained the same throughout the studies), these are included and highlighted in red font.  However, we chose to not include the dosage if (1) patients were on combination therapy (Diuretics, ACE inhibitors, ARBs, etc), (2) were subjected to washout periods during the study, and (3) where the dosage was changed before and during the study.

Pages 3,4,5,6 and 8 (of the clean copy) provide the dosages.

2-Apart from this, it is well-known that PCSK9 synergizes with pro-inflammatory pathways that, in the end, contribute to the generation of ROS and oxidized forms of LDL-C. In addition, the authors should consider the contribution of the different polymorphisms in the human PCSK9 and the gain/loss of activity as an unmet determinant in the future development of PCSK9 targeting molecules.

This is now addressed in the revised paper. Thank you for this recommendation.

Page 5:

The SLCO1B1 gene locus occupies 109 kb on chromosome 12 (Chr 12p12.2).  The common c.521T>C variant rs4149056 has been associated with decreased clearance for a number of drugs in vivo.  This variant is the most widely recognized as contributing significantly to statin myopathy [17].  All statins are transported by at least one of the known adenosine triphosphate (ATP)–binding cassette and solute carrier (SLC) transporters.  Multiple isoforms of the cytochrome P450 (CYP450) enzymes are involved in oxidative metabolism of statins, with many also undergoing conjugation via uridine 5′-diphospho-glucuronosyltransferase (UGT) enzymes.  Variations in an individual enzyme functions seem to have little effect on statin metabolism, suggesting that these configurations in combination account for the variation in response and myopathy reported by statin users [18]. Replication studies in a small population have identified three SNPs, rs9342288, rs1337512 and rs3857532, in the eyes shut homolog (EYS) on chromosome 6 suggestive of an association with the risk for severe statin myopathy [19,20].

Pages 8 and 9:

Pharmacogenomic polymorphisms appear to have dramatic influences on not only baseline lipid levels and cardiovascular risk, but some also impact response to statins in lipid reduction. The role of apoE is the regulation of lipid metabolism and involvement in the development of CVD is well-established. In apoE, the rs7412 mutation was strongly associated with LDL-C response [41]. SLCO1B1 c.521T>C variant rs4149056 does not appear to impact statin response [17]. The effect of the EYS rs9342288, rs1337512 and rs3857532, on statin efficacy is poorly documented, likely due to the severity of side effects [16,42].  PCSK9 E670G polymorphism is an independent determinant of plasma LDL-C levels, the severity of coronary atherosclerosis and is associated with severity of intracranial atherosclerosis, thus can be used as a predictor of large-vessel atherosclerosis [43,44].  Mutations in this gene have been associated with hypolipidemia through ‘‘loss-of-function’’ and hyperlipidemia through ‘‘gain-of-function’’ mechanisms.  However, it does not appear to have any influence in response to either atorvastatin or pravastatin [45,46], and like mutations in PCSK9 I474V, it has not been shown to be associated with lipid levels or CHD risk in healthy men [45].  PCSK9 Y142X and C679X are nonsense mutations associated with a lower LDL-C level, and a 40 percent reduction in mean LDL cholesterol for black patients with the polymorphisms.  These mutations are very rare in white subjects [47]. While a stop codon usually results in production of a non-functional protein, consideration should be given that this is a beneficial mutation worthy of further study.

The PCSK9 R46L polymorphism is associated with a lower LDL-C level and significantly lower LDL-C in response to atorvastatin and pravastatin.  The reduction in LDL-C and CHD has been shown and confirmed, and in fact, reduction in risk of CHD was more than predicted by reduction in LDL-C alone.  Interestingly, R46L does not appear to alter response to rosuvastatin [45].  In one study, mean intima–media thickness was slightly but significantly lower among carriers of R46L mutations [47]. Rosuvastatin increased plasma concentration of PCSK9 in proportion to the magnitude of LDL-C reduction [48], which defies expected results of increased levels of PCSK9. This requires further study to determine the mechanism of this phenomenon. 

3-Finally, a comment on combined therapy with low doses of the different cholesterol-reducing drugs/approaches deserves attention, at least from an academic and medical point of view.

We thank you for this comment. This is now reflected in the conclusion of the paper.

Revised Section, page 20:

While PCSK9 inhibitors were the primary focus of this review, it is important to give recognition to the previously mentioned statin alternative medications (ezetimibe, bempedoic acid, red yeast rice and inclisiran) as worthy candidates for investigation in future studies.  With the prevalence of patients with intolerance or resistance to statin medications, clinicians require statin-alternatives for their patients.  Fortunately, the future of LLT’s, both as a mono or dual therapy shows promise for physicians and patients alike given the growing number of options becoming available. 

Round 2

Reviewer 1 Report

The manuscript is improved following revisions based on reviewer comments. Only minor comments:

Page 5: Classification of HF into HFrEF, HFmrEF, HFimpEF and HFpEF is not exlusively a classification for non-ischemic HF.

Page 6: 'HFpEF and HFrEF heart failure' - suggest remove 'heart failure' which is redundant.

Author Response

Comments from R1.

The manuscript is improved following revisions based on reviewer comments. Only minor comments:

Page 5: Classification of HF into HFrEF, HFmrEF, HFimpEF and HFpEF is not exclusively a classification for non-ischemic HF.

Page 6: 'HFpEF and HFrEF heart failure' - suggest remove 'heart failure' which is redundant.

We have addressed these minor changes. On behalf of all my young co-authors who are completing their medical school studies, thank you for your input.
